statistical physics/mathematical modelling

Repeated games, zero-determinant strategies, memory-*n* strategies

**Author for correspondence:**
Masahiko Ueda
e-mail: m.ueda@yamaguchi-u.ac.jp

# Memory-two zero-determinant strategies in repeated games

## Masahiko Ueda

Graduate School of Sciences and Technology for Innovation, Yamaguchi University, Yamaguchi 753-8511, Japan

MU, 0000-0002-8168-0102

Repeated games have provided an explanation of how mutual cooperation can be achieved even if defection is more favourable in a one-shot game in the Prisoner's Dilemma situation. Recently found zero-determinant (ZD) strategies have substantially been investigated in evolutionary game theory. The original memory-one ZD strategies unilaterally enforce linear relationships between average pay-offs of players. Here, we extend the concept of ZD strategies to memory-two strategies in repeated games. Memory-two ZD strategies unilaterally enforce linear relationships between correlation functions of pay-offs and pay-offs of the previous round. Examples of memory-two ZD strategy in the repeated Prisoner's Dilemma game are provided, some of which generalize the tit-for-tat strategy to a memory-two case. Extension of ZD strategies to memory-*n* case with $n \geq 2$ is also straightforward.

## 1. Introduction

Repeated games offer a framework explaining forward-looking behaviours and reciprocity of rational agents [1,2]. Since it was pointed out that game theory of rational agents can be applied to evolutionary behaviour of populations of biological systems [3], evolutionary game theory has investigated the condition where mutualism is maintained in conflicts [4–11]. In the repeated Prisoner's Dilemma game, it was found that, although there are many equilibria, none of them are evolutionary stable owing to neutral drift [12]. Because rationality of each biological individual is bounded, evolutionary stability of strategies whose length of memory is one has mainly been focused on in evolutionary game theory. However, memory-one strategies contain several useful strategies in the Prisoner's Dilemma game, such as the grim trigger strategy [13], the tit-for-tat (TFT) strategy [14–17], and the win-stay lose-shift strategy [5], which can form cooperative Nash equilibria.

In 2012, Press and Dyson discovered a novel class of memory-one strategies called zero-determinant (ZD) strategies [18]. ZD strategy unilaterally enforces a linear relationships between average pay-offs of players. ZD strategies in the Prisoner's Dilemma game contain the equalizer strategy which unilaterally sets the average pay-off of the opponent, and the extortionate strategy by which the player can gain greater average pay-off than the opponent. After their work, evolutionary stability of ZD strategies in the Prisoner's Dilemma game was investigated by several authors [19–24]. Furthermore, the concept of ZD strategies has been extended to multi-player multi-action games [25–29]. Linera algebraic properties of ZD strategies in general multi-player multi-action games with many ZD players were also investigated in [30], which found that possible ZD strategies are constrained by the consistency of the linear pay-off relationships. Another extension is ZD strategies in repeated games with imperfect monitoring [30–32], where possible ZD strategies are more restricted than ones in perfect monitoring cases. Furthermore, ZD strategies were also extended to repeated games with discounting factor [28,33–35] and asymmetric games [36]. Performance of ZD strategies such as the extortionate strategy and the generous ZD strategy in the Prisoner's Dilemma game has also been investigated in human experiments [37–39]. Moreover, behaviour of the extortionate strategy in structured populations was found to be quite different from that in well-mixed populations [40,41]. Although ZD strategies are not necessarily a rational strategy, they contain the TFT strategy in the Prisoner's Dilemma game [18], which returns the opponent's previous action, and accordingly ZD strategies form a significant class of memory-one strategies.

Recently, properties of longer memory strategies have been investigated in the context of repeated games with implementation errors [42–45]. In general, longer memory enables complicated behaviour [46]. Especially, it has been shown that, in the Prisoner's Dilemma game, a memory-two strategy called tit-for-tat-anti-tit-for-tat (TFT-ATFT) is successful under implementation errors [42]. Although TFT-ATFT normally behaves as TFT, it switches to ATFT when it recognizes an error, and then returns to TFT when mutual cooperation is achieved or when the opponent unilaterally defects twice. In [45], a successful strategy in memory-three class which can easily be interpreted has also been proposed. Recall that original memory-one TFT strategy, which is also successful but is not robust against errors, is a special case of memory-one ZD strategies. Therefore, discussion of longer-memory strategies in the context of ZD strategies would be useful. However, the concept of ZD strategies has not been extended to longer-memory strategies.

In this paper, we extend the concept of ZD strategies to memory-two strategies in repeated games. Memory-two ZD strategies unilaterally enforce linear relationships between correlation functions of pay-offs at the present round and pay-offs at the previous round. We provide examples of memory-two ZD strategy in a repeated Prisoner's Dilemma game. Particularly, one of the examples can be regarded as an extension of TFT strategy to a memory-two case.

The paper is organized as follows. In §2, we introduce a model of repeated game with memory-two strategies. In §3, we extend ZD strategies to the memory-two strategy class. In §4, we provide examples of memory-two ZD strategies in a repeated Prisoner's Dilemma game. In §5, extension of ZD strategies to a memory-$n$ case with $n \geq 2$ is discussed. Section 6 is devoted to concluding remarks.

## 2. Model

We consider an $N$-player repeated game. Action of player $a \in \{1,\ldots, N\}$ is described as $\sigma_a \in \{1,\ldots, M\}$. We collectively denote state $\sigma := (\sigma_1, \ldots, \sigma_N)$. We consider the situation that the length of memory of strategies of all players are at most two. (We will see in §6 that this assumption can be weakened to memory-$n$ with $n \geq 2$.) Strategy of player $a$ is described as the conditional probability $T_a(\sigma_a|\sigma', \sigma'')$ of taking acton $\sigma_a$ when states at last round and second-to-last round are $\sigma'$ and $\sigma''$, respectively. Let $s_a(\sigma)$ be pay-offs of player $a$ when the state is $\sigma$. The time evolution of this system is described by the Markov chain

$$P(\sigma, \sigma', t+1) = \sum_{\sigma''} T(\sigma|\sigma', \sigma'')P(\sigma', \sigma'', t), \tag{2.1}$$

where $P(\sigma, \sigma', t)$ is joint distribution of the present state $\sigma$ and the last state $\sigma'$ at time $t$, and we have defined the transition probability

$$T(\sigma|\sigma', \sigma'') := \prod_{a=1}^{N} T_a(\sigma_a|\sigma', \sigma''). \tag{2.2}$$

Initial condition is described as $P_0(\sigma, \sigma')$. We consider the situation that the discounting factor is $\delta = 1$ [1].

# 3. Memory-two zero-determinant strategies

We consider the situation that the Markov chain (equation (2.1)) has a stationary probability distribution:

$$P^{(st)}(\sigma, \sigma') = \sum_{\sigma''} T(\sigma|\sigma', \sigma'')P^{(st)}(\sigma', \sigma'').  \tag{3.1}$$

By taking summation of both sides with respect to $\sigma_{-a} := \sigma \backslash \sigma_a$ with an arbitrary $a$, we obtain

$$0 = \sum_{\sigma''} T_a(\sigma_a|\sigma', \sigma'')P^{(st)}(\sigma', \sigma'') - \sum_{\sigma''} \delta_{\sigma_a,\sigma_a''}P^{(st)}(\sigma'', \sigma').  \tag{3.2}$$

Furthermore, by taking summation of both sides with respective to $\sigma'$, we obtain

$$0 = \sum_{\sigma'}\sum_{\sigma''}[T_a(\sigma_a|\sigma', \sigma'') - \delta_{\sigma_a,\sigma_a'}]P^{(st)}(\sigma', \sigma'').  \tag{3.3}$$

Therefore, the quantity

$$\hat{T}_a(\sigma_a|\sigma', \sigma'') := T_a(\sigma_a|\sigma', \sigma'') - \delta_{\sigma_a,\sigma_a'}  \tag{3.4}$$

is mean-zero with respective to the stationary distribution $P^{(st)}(\sigma', \sigma'')$:

$$0 = \sum_{\sigma'}\sum_{\sigma''}\hat{T}_a(\sigma_a|\sigma', \sigma'')P^{(st)}(\sigma', \sigma'')  \tag{3.5}$$

for arbitrary $\sigma_a$. This is the extension of Akin's lemma [25,28,30,47,48] to the memory-two case. (We remark that the term $\delta_{\sigma_a,\sigma_a'}$ is regarded as memory-one strategy 'Repeat', which repeats the action at the previous round.) We call $\hat{T}_a(\sigma_a) := (\hat{T}_a(\sigma_a|\sigma', \sigma''))$ as a Press–Dyson (PD) matrix. It should be noted that $\hat{T}_a(\sigma_a)$ is controlled only by player $a$.

When player $a$ chooses her strategies as her PD matrices satisfy

$$\sum_{\sigma_a} c_{\sigma_a}\hat{T}_a(\sigma_a|\sigma', \sigma'') = \sum_{b=0}^{N}\sum_{c=0}^{N}\alpha_{b,c}s_b(\sigma')s_c(\sigma''),  \tag{3.6}$$

with some coefficients $\{c_{\sigma_a}\}$ and $\{\alpha_{b,c}\}$, where we have introduced $s_0(\sigma) := 1$, we obtain

$$\begin{aligned}0 &= \sum_{\sigma'}\sum_{\sigma''}\left[\sum_{b=0}^{N}\sum_{c=0}^{N}\alpha_{b,c}s_b(\sigma')s_c(\sigma'')\right]P^{(st)}(\sigma', \sigma'')\\ &= \sum_{b=0}^{N}\sum_{c=0}^{N}\alpha_{b,c}\langle s_b(\sigma(t+1))s_c(\sigma(t))\rangle^{(st)},\end{aligned}  \tag{3.7}$$

where $\langle \cdots \rangle^{(st)}$ represents average with respect to the stationary distribution $P^{(st)}$. This is the extension of the concept of ZD strategies to a memory-two case. We remark that the original (memory-one) ZD strategies unilaterally enforce linear relationships between average pay-offs of players at the stationary state. Here, memory-two ZD strategies unilaterally enforce linear relationships between correlation functions of pay-offs at the present round and pay-offs at the previous round at the stationary state. (It should be noted that the quantity $\langle s_b(\sigma(t+1))s_c(\sigma(t))\rangle^{(st)}$ does not depend on $t$ in the stationary state.) We note that because the number of the components of a PD matrix is $M^{2N}$ and the number of pay-off tensors $s_b \otimes s_c$ in equation (3.6) is $(N+1)^2$, the space of memory-two ZD strategies is small even for the Prisoner's Dilemma game ($N = 2$ and $M = 2$), and most of the memory-two strategies are not memory-two ZD strategies. In addition, although we choose $s_b \otimes s_c$ as a basis in the right-hand side of equation (3.6), such choice is not necessary and we can choose another function [49].

We remark that, when we take summation of both sides of equation (3.1) with respect to $\sigma$, we obtain

$$\sum_{\sigma} P^{(st)}(\sigma, \sigma') = \sum_{\sigma''} P^{(st)}(\sigma', \sigma'').  \tag{3.8}$$

This uniquely determines the stationary distribution of a single state $\sigma'$.

We also remark that, because of the normalization condition of the conditional probability $T_a(\sigma_a|\sigma', \sigma'')$, PD matrices satisfy

$$\sum_{\sigma_a} \hat{T}_a(\sigma_a|\,\sigma',\,\sigma'') = 0 \tag{3.9}$$

for arbitrary $(\sigma', \sigma'')$.

# 4. Examples: repeated Prisoner's Dilemma

Here, we consider a two-player two-action Prisoner's Dilemma game [18]. Actions of two players are 1 (cooperation) or 2 (defection). Pay-offs of two players $s_a := (s_a(\sigma))$ are $s_1 = (R, S, T, P)$ and $s_2 = (R, T, S, P)$ with $T > R > P > S$. We provide three examples of memory-two ZD strategies.

## 4.1. Example 1: relating correlation function with average pay-offs

We consider the situation that player 1 takes the following memory-two strategy:

$$
\begin{aligned}
T_1(1) := & \begin{pmatrix}
T_1(1|1,1,1,1) & T_1(1|1,1,1,2) & T_1(1|1,1,2,1) & T_1(1|1,1,2,2) \\
T_1(1|1,2,1,1) & T_1(1|1,2,1,2) & T_1(1|1,2,2,1) & T_1(1|1,2,2,2) \\
T_1(1|2,1,1,1) & T_1(1|2,1,1,2) & T_1(1|2,1,2,1) & T_1(1|2,1,2,2) \\
T_1(1|2,2,1,1) & T_1(1|2,2,1,2) & T_1(1|2,2,2,1) & T_1(1|2,2,2,2)
\end{pmatrix} \\
= & \begin{pmatrix}
1 - \frac{(R-P)(R-S)}{(T-P)(T-S)} & 1 & 1 - \frac{R-P}{T-P} & 1 - \frac{(R-P)(P-S)}{(T-P)(T-S)} \\
1 - \frac{R-S}{T-S} & 1 & 0 & 1 - \frac{P-S}{T-S} \\
\frac{(P-S)(R-S)}{(T-P)(T-S)} & 0 & \frac{P-S}{T-P} & \frac{(P-S)^2}{(T-P)(T-S)} \\
0 & 0 & 0 & 0
\end{pmatrix}.
\end{aligned} \tag{4.1}
$$

(We have assumed that $T - P \geq P - S$. For the case $T - P < P - S$, a slight modification is needed.) Then, we find that her PD matrix is

$$
\begin{aligned}
\hat{T}_1(1) = & \begin{pmatrix}
T_1(1|1,1,1,1) - 1 & T_1(1|1,1,1,2) - 1 & T_1(1|1,1,2,1) - 1 & T_1(1|1,1,2,2) - 1 \\
T_1(1|1,2,1,1) - 1 & T_1(1|1,2,1,2) - 1 & T_1(1|1,2,2,1) - 1 & T_1(1|1,2,2,2) - 1 \\
T_1(1|2,1,1,1) & T_1(1|2,1,1,2) & T_1(1|2,1,2,1) & T_1(1|2,1,2,2) \\
T_1(1|2,2,1,1) & T_1(1|2,2,1,2) & T_1(1|2,2,2,1) & T_1(1|2,2,2,2)
\end{pmatrix} \\
= & \begin{pmatrix}
-\frac{(R-P)(R-S)}{(T-P)(T-S)} & 0 & -\frac{R-P}{T-P} & -\frac{(R-P)(P-S)}{(T-P)(T-S)} \\
-\frac{R-S}{T-S} & 0 & -1 & -\frac{P-S}{T-S} \\
\frac{(P-S)(R-S)}{(T-P)(T-S)} & 0 & \frac{P-S}{T-P} & \frac{(P-S)^2}{(T-P)(T-S)} \\
0 & 0 & 0 & 0
\end{pmatrix},
\end{aligned} \tag{4.2}
$$

which means

$$\hat{T}_1(1|\,\sigma',\,\sigma'') = -\frac{1}{(T-P)(T-S)}[s_2(\sigma') - P][s_1(\sigma'') - S], \tag{4.3}$$

and that this strategy is a memory-two ZD strategy which unilaterally enforces

$$0 = \langle s_2(\sigma(t+1))s_1(\sigma(t))\rangle^{(\text{st})} - S\langle s_2\rangle^{(\text{st})} - P\langle s_1\rangle^{(\text{st})} + PS. \tag{4.4}$$

Therefore, the correlation function $\langle s_2(\sigma(t+1))s_1(\sigma(t))\rangle^{(\text{st})}$ is related to the average pay-offs $\langle s_1\rangle^{(\text{st})}$ and $\langle s_2\rangle^{(\text{st})}$ by the ZD strategy.

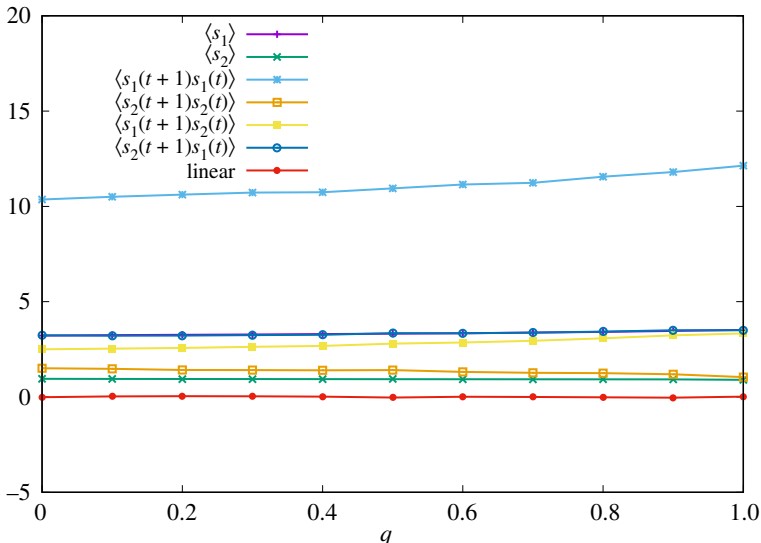

**Figure 1.** Time-averaged pay-offs of two players $\sum_{t'=1}^{t} s_a(\sigma(t'))/t$ and correlation functions $\sum_{t'=1}^{t} s_a(\sigma(t'))s_b(\sigma(t'-1))/t$ with $t = 100\,000$ for various $q$ when the strategy of player 1 is given by equation (4.1). The red line corresponds to the right-hand side of equation (4.4).

We provide numerical results about this linear relationship. We set parameters $(R, S, T, P) = (3, 0, 5, 1)$. The strategy of player 2 is set to

$$
T_2(1) := \begin{pmatrix} T_2(1|1,1,1,1) & T_2(1|1,1,1,2) & T_2(1|1,1,2,1) & T_2(1|1,1,2,2) \\ T_2(1|1,2,1,1) & T_2(1|1,2,1,2) & T_2(1|1,2,2,1) & T_2(1|1,2,2,2) \\ T_2(1|2,1,1,1) & T_2(1|2,1,1,2) & T_2(1|2,1,2,1) & T_2(1|2,1,2,2) \\ T_2(1|2,2,1,1) & T_2(1|2,2,1,2) & T_2(1|2,2,2,1) & T_2(1|2,2,2,2) \end{pmatrix}
$$

$$
= \begin{pmatrix} q & q & q & q \\ \frac{2}{3} & \frac{2}{3} & \frac{2}{3} & \frac{2}{3} \\ \frac{2}{3} & \frac{2}{3} & \frac{2}{3} & \frac{2}{3} \\ \frac{2}{3} & \frac{2}{3} & \frac{2}{3} & \frac{2}{3} \end{pmatrix} \tag{4.5}
$$

and we change $q$ in the range $[0, 1]$. (Note that the strategy of player 2 is essentially memory-one.) Actions of both players at $t = 0$ and $t = 1$ are sampled from uniform distribution. In figure 1, we display the result of numerical simulations of one sample, where the average is calculated by time average at $t = 100000$. We can see that the linear relation (4.4) indeed holds for all $q$.

## 4.2. Example 2: extended tit-for-tat strategy

Here, we introduce a memory-two ZD strategy which can be called an extended tit-for-tat (ETFT) strategy. We consider the situation when player 1 takes the following memory-two strategy:

$$
T_1(1) = \begin{pmatrix} 1 & 1 & 1 & 1 \\ \frac{1}{2} & 0 & 1 & \frac{1}{2} \\ \frac{1}{2} & 1 & 0 & \frac{1}{2} \\ 0 & 0 & 0 & 0 \end{pmatrix}. \tag{4.6}
$$

Note that this does not depend on the concrete values of pay-offs $(R, S, T, P)$. Then, we find that her PD matrix is

$$
\hat{T}_1(1) = \begin{pmatrix} 0 & 0 & 0 & 0 \\ -\frac{1}{2} & -1 & 0 & -\frac{1}{2} \\ \frac{1}{2} & 1 & 0 & \frac{1}{2} \\ 0 & 0 & 0 & 0 \end{pmatrix} \tag{4.7}
$$

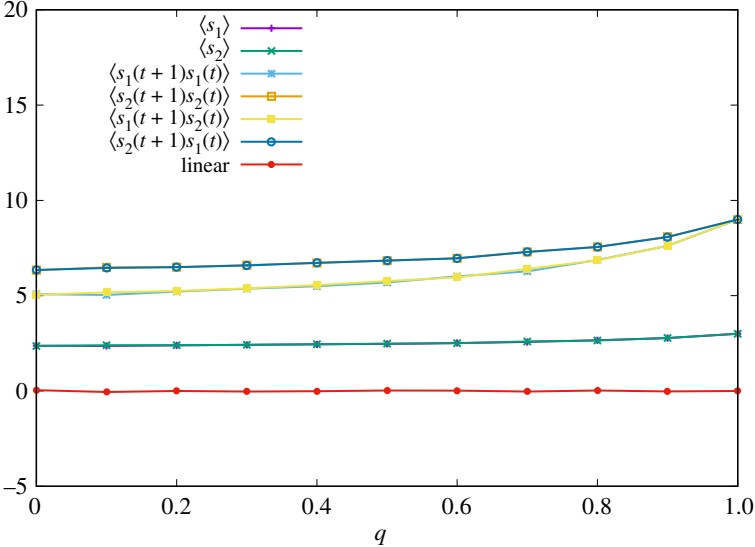

**Figure 2.** Time-averaged pay-offs of two players $\sum_{t'=1}^{t} s_a(\sigma(t'))/t$ and correlation functions $\sum_{t'=1}^{t} s_a(\sigma(t'))s_b(\sigma(t'-1))/t$ with $t = 100\,000$ for various $q$ when the strategy of player 1 is given by equation (4.6). The red line corresponds to the right-hand side of equation (4.9).

and it satisfies

$$\hat{T}_1(1|\,\sigma',\,\sigma'') = \frac{1}{2(T-S)^2}[\{s_1(\sigma') - s_2(\sigma')\}\{s_2(\sigma'') - s_1(\sigma'')\} + (T-S)\{s_1(\sigma') - s_2(\sigma')\}]. \qquad (4.8)$$

Therefore, it is memory-two ZD strategy, which enforces

$$
\begin{aligned}
0 ={}& \langle s_1(\sigma(t+1))s_2(\sigma(t))\rangle^{(\mathrm{st})} + \langle s_2(\sigma(t+1))s_1(\sigma(t))\rangle^{(\mathrm{st})} \\
& - \langle s_1(\sigma(t+1))s_1(\sigma(t))\rangle^{(\mathrm{st})} - \langle s_2(\sigma(t+1))s_2(\sigma(t))\rangle^{(\mathrm{st})} \\
& + (T-S)\Big\{\langle s_1\rangle^{(\mathrm{st})} - \langle s_2\rangle^{(\mathrm{st})}\Big\}.
\end{aligned}
\qquad (4.9)
$$

This linear relationship can be seen as some fairness condition between two players. Because the original TFT strategy

$$T_1(1) = \begin{pmatrix} 1 & 1 & 1 & 1 \\ 0 & 0 & 0 & 0 \\ 1 & 1 & 1 & 1 \\ 0 & 0 & 0 & 0 \end{pmatrix} \qquad (4.10)$$

enforces $\langle s_1\rangle^{(\mathrm{st})} = \langle s_2\rangle^{(\mathrm{st})}$ [18], equation (4.6) can be regarded as an extension of TFT strategy.

We provide numerical results about the ETFT strategy. Parameters and strategies of player 2 are set to the same values as those in the previous subsection. In figure 2, we display the result of numerical simulations of one sample. We can check that the linear relationship of equation (4.9) seems to hold for all $q$.

We can understand feelings of the ETFT player as follows. We find that, when the previous state is $(1, 1)$ or $(2, 2)$, ETFT behaves as TFT. When the previous state and the second-to-last state are $(\sigma', \sigma'') = ((1, 2), (1, 1))$ or $((2, 1), (1, 1))$, ETFT regards that one of the players mistakes his/her action, and returns action 1 (cooperation) or 2 (defection) at random. Similarly, when the previous state and the second-to-last state are $((1, 2), (2, 2))$ or $((2, 1), (2, 2))$, ETFT also regards that one of the players mistakes his/her action, and returns action 1 or 2 at random. When the previous state and the second-to-last state are $((1, 2), (1, 2))$, ETFT ceases to cooperate and returns action 2. When the previous state and the second-to-last state are $((2, 1), (2, 1))$, ETFT continues to exploit and returns action 2. Finally, when the previous state and the second-to-last state are $((1, 2), (2, 1))$ or $((2, 1), (1, 2))$, ETFT generously

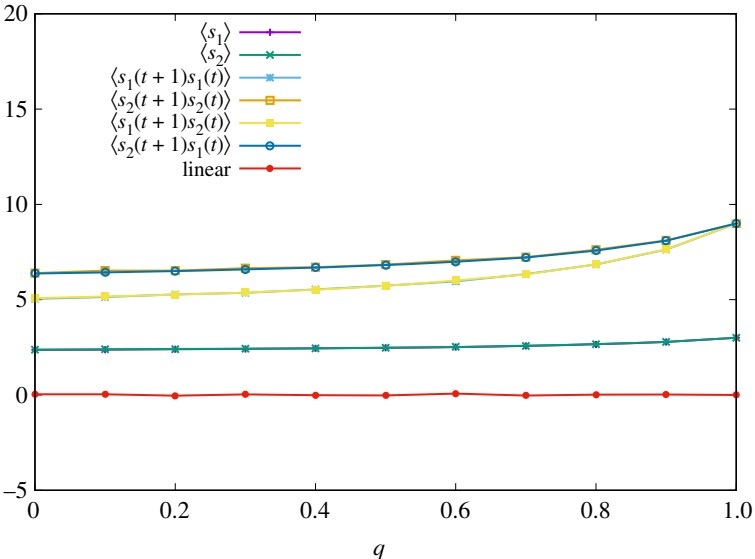

**Figure 3.** Time-averaged pay-offs of two players $\sum_{t'=1}^{t} s_a(\sigma(t'))/t$ and correlation functions $\sum_{t'=1}^{t} s_a(\sigma(t'))s_b(\sigma(t'-1))/t$ with $t = 100\,000$ for various $q$ when the strategy of player 1 is given by equation (4.12). The red line corresponds to the right-hand side of equation (4.14).

cooperates. Although this strategy is different from TFT-ATFT [42]:

$$T_1(1) = \begin{pmatrix} 1 & 1 & 1 & 1 \\ 0 & 0 & 0 & 1 \\ 0 & 1 & 0 & 1 \\ 1 & 0 & 1 & 0 \end{pmatrix}, \tag{4.11}$$

which is deterministic, ETFT may be successful because ETFT has several properties in common with TFT-ATFT. Furthermore, because ETFT is stochastic, it may be robust against implementation errors. Evolutionary stability of ETFT must be investigated in future.

It should be noted that a slightly modified version of ETFT:

$$T_1(1) = \begin{pmatrix} 1 & 1 & 1 & 1 \\ \frac{1}{2} & 1 & 0 & \frac{1}{2} \\ \frac{1}{2} & 0 & 1 & \frac{1}{2} \\ 0 & 0 & 0 & 0 \end{pmatrix} \tag{4.12}$$

is also a memory-two ZD strategy. We call this strategy as type-2 extended tit-for-tat (ETFT-2) strategy. The PD matrix of ETFT-2 is described as

$$\hat{T}_1(1 \mid \sigma', \sigma'') = -\frac{1}{2(T-S)^2} [\{s_1(\sigma') - s_2(\sigma')\}\{s_2(\sigma'') - s_1(\sigma'')\} - (T-S)\{s_1(\sigma') - s_2(\sigma')\}], \tag{4.13}$$

which enforces a linear relationship

$$\begin{aligned} 0 = {}& \langle s_1(\sigma(t+1))s_2(\sigma(t))\rangle^{(\mathrm{st})} + \langle s_2(\sigma(t+1))s_1(\sigma(t))\rangle^{(\mathrm{st})} \\ & - \langle s_1(\sigma(t+1))s_1(\sigma(t))\rangle^{(\mathrm{st})} - \langle s_2(\sigma(t+1))s_2(\sigma(t))\rangle^{(\mathrm{st})} \\ & - (T-S)\left\{\langle s_1\rangle^{(\mathrm{st})} - \langle s_2\rangle^{(\mathrm{st})}\right\}. \end{aligned} \tag{4.14}$$

That is, the sign of the last term is different from that of ETFT. In figure 3, we also display the result of numerical simulations of one sample, where parameters are set to the same values as before. We can check that equation (4.14) holds for all $q$. Although ETFT-2 is similar to ETFT, it will be exploited by all-$D$ strategy (which always defects), because $T_1(1 \mid 1, 2, 1, 2) = 1$. Therefore, it is expected that ETFT-2 is less successful than ETFT.

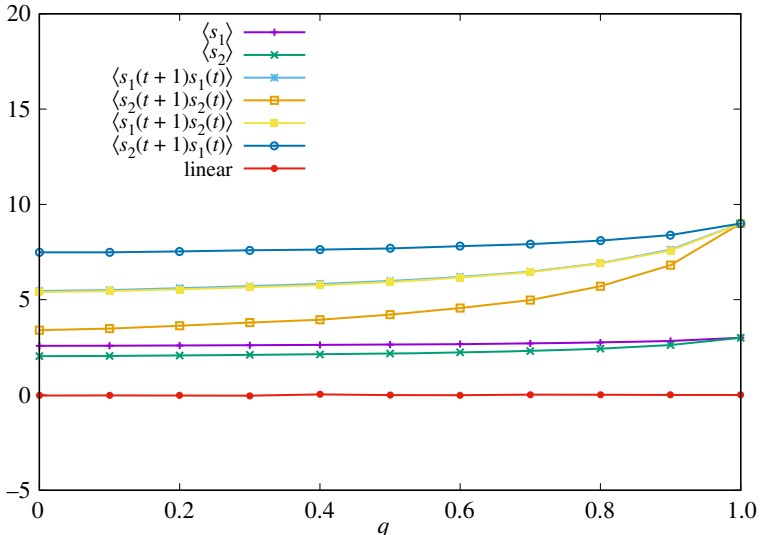

**Figure 4.** Time-averaged pay-offs of two players $\sum_{t'=1}^{t} s_a(\sigma(t'))/t$ and correlation functions $\sum_{t'=1}^{t} s_a(\sigma(t'))s_b(\sigma(t'-1))/t$ with $t = 100\,000$ for various $q$ when the strategy of player 1 is given by equation (4.15). The red line corresponds to the right-hand side of equation (4.18).

## 4.3. Example 3: fickle tit-for-tat strategy

Here, we introduce another memory-two ZD strategy which can be called a fickle tit-for-tat (FTFT) strategy. In this subsection, we assume $2R > T + S$, which corresponds to the condition that mutual cooperation is favourable to the period-two sequence $(1, 2) \to (2, 1) \to (1, 2) \to \cdots$ [18]. We consider the situation that player 1 takes the following memory-two strategy:

$$T_1(1) = \begin{pmatrix} 1 & 1 & 1 & 1 \\ 0 & 1 - \frac{T+S}{2R} & 1 - \frac{T+S}{2R} & 1 - \frac{P}{R} \\ 1 & \frac{T+S}{2R} & \frac{T+S}{2R} & \frac{P}{R} \\ 0 & 0 & 0 & 0 \end{pmatrix}. \tag{4.15}$$

Then, we find that her PD matrix is

$$\hat{T}_1(1) = \begin{pmatrix} 0 & 0 & 0 & 0 \\ -1 & -\frac{T+S}{2R} & -\frac{T+S}{2R} & -\frac{P}{R} \\ 1 & \frac{T+S}{2R} & \frac{T+S}{2R} & \frac{P}{R} \\ 0 & 0 & 0 & 0 \end{pmatrix} \tag{4.16}$$

and it satisfies

$$\hat{T}_1(1|\sigma', \sigma'') = \frac{1}{2R(T-S)}[s_1(\sigma')s_2(\sigma'') - s_2(\sigma')s_1(\sigma'') + s_1(\sigma')s_1(\sigma'') - s_2(\sigma')s_2(\sigma'')]. \tag{4.17}$$

Therefore, it is also memory-two ZD strategy, which enforces

$$\begin{aligned} 0 = {}& \langle s_1(\sigma(t+1))s_2(\sigma(t)) \rangle^{(\text{st})} - \langle s_2(\sigma(t+1))s_1(\sigma(t)) \rangle^{(\text{st})} \\ & + \langle s_1(\sigma(t+1))s_1(\sigma(t)) \rangle^{(\text{st})} - \langle s_2(\sigma(t+1))s_2(\sigma(t)) \rangle^{(\text{st})}. \end{aligned} \tag{4.18}$$

This linear relationship can be regarded as another type of fairness condition between two players. One can compare the strategy matrix of FTFT (equation (4.15)) with that of TFT (equation (4.10)). FTFT can take a different action from TFT with finite probability when the previous state is $(1, 2)$ or $(2, 1)$.

We provide numerical results about the FTFT strategy. Parameters and strategies of player 2 are set to the same values as those in the previous subsections. In figure 4, we display the result of numerical simulations of one sample. We confirm that the linear relationship equation (4.18) holds for all $q$.

## 4.4. Example 4: extended zero-sum strategy

Here we assume that $2R > T + S$ and $2P < T + S$. In memory-one strategies, there exists the following memory-one ZD strategy:

$$
T_1(1) = \begin{pmatrix}
1 - \frac{2R-(T+S)}{A} & 1 - \frac{2R-(T+S)}{A} & 1 - \frac{2R-(T+S)}{A} & 1 - \frac{2R-(T+S)}{A} \\
1 & 1 & 1 & 1 \\
0 & 0 & 0 & 0 \\
\frac{(T+S)-2P}{A} & \frac{(T+S)-2P}{A} & \frac{(T+S)-2P}{A} & \frac{(T+S)-2P}{A}
\end{pmatrix}, \tag{4.19}
$$

where we have introduced

$$
A := \max\ \{2R - (T+S), (T+S) - 2P\}. \tag{4.20}
$$

Because this strategy satisfies

$$
\hat{T}_1(1|\,\sigma',\,\sigma'') = -\frac{1}{A}\{s_1(\sigma') + s_2(\sigma') - (T+S)\}, \tag{4.21}
$$

it unilaterally enforces

$$
0 = \langle s_1 \rangle^{(st)} + \langle s_2 \rangle^{(st)} - (T+S). \tag{4.22}
$$

Because this relationship means that the sum of average pay-offs of two players is fixed, this ZD strategy can be called a zero-sum strategy (ZSS).

As an extension of ZSS, we can consider the following memory-two strategy:

$$
T_1(1) = \begin{pmatrix}
1 - \frac{2R-(T+S)}{A} & 1 - \frac{T+S}{2R}\frac{2R-(T+S)}{A} & 1 - \frac{T+S}{2R}\frac{2R-(T+S)}{A} & 1 - \frac{P}{R}\frac{2R-(T+S)}{A} \\
1 & 1 & 1 & 1 \\
0 & 0 & 0 & 0 \\
\frac{(T+S)-2P}{A} & \frac{T+S}{2R}\frac{(T+S)-2P}{A} & \frac{T+S}{2R}\frac{(T+S)-2P}{A} & \frac{P}{R}\frac{(T+S)-2P}{A}
\end{pmatrix}. \tag{4.23}
$$

Then, we find that a PD matrix of the player can be rewritten as

$$
\hat{T}_1(1|\,\sigma',\,\sigma'') = -\frac{1}{2RA}\{s_1(\sigma') + s_2(\sigma') - (T+S)\}\{s_1(\sigma'') + s_2(\sigma'')\}. \tag{4.24}
$$

Therefore, this strategy is a memory-two ZD strategy enforcing

$$
\begin{aligned}
0 = {} & \langle s_1(\sigma(t+1))s_1(\sigma(t))\rangle^{(st)} + \langle s_2(\sigma(t+1))s_2(\sigma(t))\rangle^{(st)} \\
& + \langle s_1(\sigma(t+1))s_2(\sigma(t))\rangle^{(st)} + \langle s_2(\sigma(t+1))s_1(\sigma(t))\rangle^{(st)} \\
& - (T+S)\left\{\langle s_1 \rangle^{(st)} + \langle s_2 \rangle^{(st)}\right\}.
\end{aligned} \tag{4.25}
$$

Because this strategy can be regarded as an extension of ZSS, we call this strategy an extended zero-sum strategy (EZSS).

In figure 5, we display the result of numerical simulation of one sample, where parameters are set to the same values as before. We can check that equation (4.25) indeed holds for all $q$.

## 4.5. Remark

As in memory-one cases [18], possible memory-two ZD strategies are restricted by the sign of each component of matrix $\hat{T}_1(1)$, because $T_1(1|\sigma',\sigma'')$ is probability for all $(\sigma',\sigma'')$ and must be $0 \leq T_1(1|\sigma',\sigma'') \leq 1$. For example, memory-two ZD strategy of player 1 satisfying

$$
\hat{T}_1(1|\,\sigma',\,\sigma'') = -\frac{1}{(T-P)(T-S)}[s_1(\sigma') - P][s_2(\sigma'') - S], \tag{4.26}
$$

does not exist.

# 5. Extension to memory-$n$ case

In this section, we discuss that extension of ZD strategies to a memory-$n$ ($n \geq 2$) case. By using the technique of this paper, extension of Akin's lemma (equation (3.5)) to a longer-memory case is straightforward. Therefore, extension of the concept of ZD strategies to longer-memory cases is also

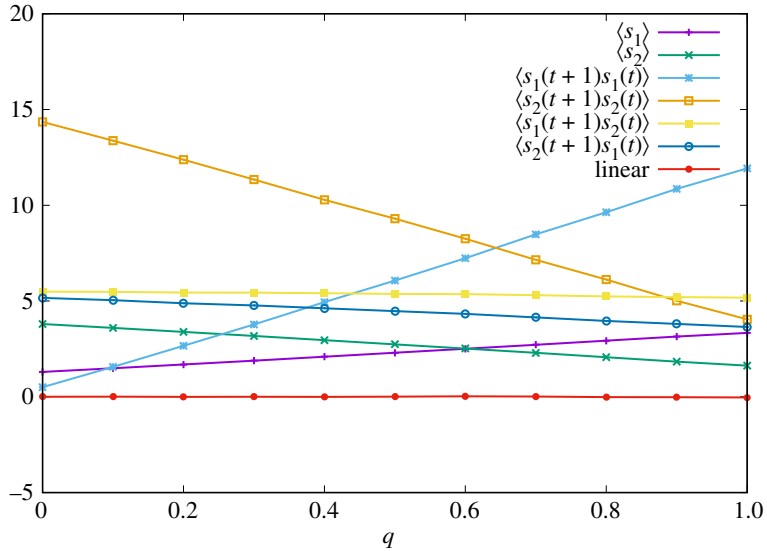

**Figure 5.** Time-averaged pay-offs of two players $\sum_{t'=1}^{t} s_a(\sigma(t'))/t$ and correlation functions $\sum_{t'=1}^{t} s_a(\sigma(t'))s_b(\sigma(t'-1))/t$ with $t = 100\,000$ for various $q$ when the strategy of player 1 is given by equation (4.23). The red line corresponds to the right-hand side of equation (4.25).

straightforward. For the memory-$n$ case, the time evolution is described by the Markov chain

$$P(\sigma, \sigma^{(-1)}, \ldots, \sigma^{(-n+1)}, t+1) = \sum_{\sigma^{(-n)}} T(\sigma|\sigma^{(-1)}, \ldots, \sigma^{(-n)})P(\sigma^{(-1)}, \ldots, \sigma^{(-n)}, t) \tag{5.1}$$

with the transition probability

$$T(\sigma|\sigma^{(-1)}, \ldots, \sigma^{(-n)}) := \prod_{a=1}^{N} T_a(\sigma_a|\sigma^{(-1)}, \ldots, \sigma^{(-n)}). \tag{5.2}$$

Then, by taking summation of both sides of the stationary condition with respect to $\sigma_{-a}, \sigma^{(-1)}, \cdots, \sigma^{(-n+1)}$, we obtain the extended Akin's lemma:

$$0 = \sum_{\sigma^{(-1)}} \cdots \sum_{\sigma^{(-n)}} \hat{T}_a(\sigma_a|\sigma^{(-1)}, \ldots, \sigma^{(-n)})P^{(\mathrm{st})}(\sigma^{(-1)}, \ldots, \sigma^{(-n)}) \tag{5.3}$$

with

$$\hat{T}_a(\sigma_a|\sigma^{(-1)}, \ldots, \sigma^{(-n)}) := T_a(\sigma_a|\sigma^{(-1)}, \ldots, \sigma^{(-n)}) - \delta_{\sigma_a,\sigma_a^{(-1)}}, \tag{5.4}$$

which can be called a PD tensor. When player $a$ chooses her strategy as her PD tensors satisfy

$$\sum_{\sigma_a} c_{\sigma_a} \hat{T}_a(\sigma_a|\sigma^{(-1)}, \ldots, \sigma^{(-n)})$$
$$= \sum_{b^{(-1)}=0}^{N} \cdots \sum_{b^{(-n)}=0}^{N} \alpha_{b^{(-1)},\ldots,b^{(-n)}} s_{b^{(-1)}}(\sigma^{(-1)}) \cdots s_{b^{(-n)}}(\sigma^{(-n)}) \tag{5.5}$$

with some coefficients $\{c_{\sigma_a}\}$ and $\{\alpha_{b^{(-1)},\ldots,b^{(-n)}}\}$, she unilaterally enforces a linear relationship

$$0 = \sum_{b^{(-1)}=0}^{N} \cdots \sum_{b^{(-n)}=0}^{N} \alpha_{b^{(-1)},\ldots,b^{(-n)}} \langle s_{b^{(-1)}}(\sigma(t+n-1)) \cdots s_{b^{(-n)}}(\sigma(t)) \rangle^{(\mathrm{st})}. \tag{5.6}$$

This is the memory-$n$ ZD strategy. In other words, in memory-$n$ ZD strategies, linear relationships between correlation functions of pay-offs during timespan $n$ are unilaterally enforced. Constructing useful examples of memory-$n$ ZD strategies with $n \geq 2$ in the Prisoner's Dilemma game is a subject of future work.

# 6. Concluding remarks

In this paper, we extended the concept of ZD strategies in repeated games to memory-two strategies. Memory-two ZD strategies unilaterally enforce linear relationships between correlation functions of pay-offs and pay-offs at the previous round. We provided examples of memory-two ZD strategies in the Prisoner's Dilemma game. Some of them can be regarded as variants of TFT strategy. We also discussed that extension of ZD strategies to a memory-$n$ ($n \geq 2$) case is straightforward.

Before ending this paper, we make two remarks. First, in our numerical simulations, we investigated only simple situations which correspond to well-mixed populations and without evolutionary behaviour. It has been known that evolutionary behaviour can drastically change when populations are structured [50–52]. Therefore, investigating performance of our variants of the TFT strategy in evolutionary game theory in well-mixed populations and structured populations is an important future problem.

The second remark is related to the length of memory. Recently, it has been found that long memory can promote cooperation in the Prisoner's Dilemma game [53,54]. Our variants of the TFT strategy may promote cooperation because they are constructed based on TFT. Furthermore, as discussed in §4, ETFT has several properties in common with TFT-ATFT [42], which is successful under implementation errors. Investigating which extension of TFT is the most successful is a significant problem. Additionally, whether TFT-ATFT is memory-two ZD strategy or not should be studied.

Data accessibility. Data have been uploaded to the Dryad Digital Repository: https://doi.org/10.5061/dryad.612jm6435 [55].

Competing interests. I declare I have no competing interests.

Funding. This study was supported by JSPS KAKENHI grant no. JP20K19884.

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
