## [Peer Review File · Royal Society Open Science]

Review History

RSOS-202186.R0 (Original submission)

Review form: Reviewer 1

Is the manuscript scientifically sound in its present form?

Yes

Are the interpretations and conclusions justified by the results?

Yes

Is the language acceptable?

Yes

Do you have any ethical concerns with this paper?

No

Have you any concerns about statistical analyses in this paper?

No

Recommendation?

Accept with minor revision (please list in comments)

Comments to the Author(s)

The author has investigated memory-zero-determinant (ZD) strategies with general $n > 1$. The finding is that such strategies enforce linear relations between correlation functions of payoffs over n time steps. Overall, the manuscript is clearly written about an interesting topic, and such understanding has great potential importance if we consider the recent development of memory strategies that overcome shortcomings of the original ZD strategies. I did not follow all the details of the calculation, but the result looks convincing enough according to the numerical calculation. I recommend publication of this manuscript. Here are some minor comments that I would like the author to consider:

1. Line #32~33: "when the opponent 'unilaterally' defects twice."
2. If we think of the conventional prisoner's dilemma game, Eq. (3.6) seems over-determined because it contains 16 equations with 11 unknowns. Is it possible to specify conditions for the solution to exist?
3. Although Eq. (3.6) clearly shows an algebraic structure of the memory-two ZD strategies, I wonder if we really need the product of s_b and s_c on the right-hand side. There may be a degree of freedom to choose another function of s_b and s_c .

Review form: Reviewer 2

Is the manuscript scientifically sound in its present form?

No

Are the interpretations and conclusions justified by the results?

Yes

Is the language acceptable?

No

Do you have any ethical concerns with this paper?

No

Have you any concerns about statistical analyses in this paper?

No

Recommendation?

Major revision is needed (please make suggestions in comments)

Comments to the Author(s)

I would like to support this work because it contains solid calculation, but I strongly believe that present version is not appropriate for a general reader of R. Soc. Open Sci. Please note that this journal has highly interdisciplinary character with broad range of readers from rather different backgrounds. Therefore it is essential to present works in a way that is reachable for an ordinary reader.

For example the intro seems to be too brief and it is hard to find out what was really known earlier and what was the proper inspiration of present calculations. For example several previous works were ignored which also discussed extortion or TFT strategy in the mentioned social

dilemma. Without giving a full list, let me just mention some representative papers: Sci. Rep. 4 (2014) 5496; Nat. Commun. 10 (2019) 783; Chaos Solitons and Fractals 141 (2020) 110421; Phys. Rev. E 80 (2009) 056104; Chaos, Solitons and Fractals 138 (2020) 109935.

It would be important to stress that present calculation assumes well-mixed populations while results may change in structured populations where interactions are fairly fixed and limited (see J. R. Soc. Interface 10 (2013) 20120997 and EPL 131 (2020) 68001).

In general I found that the figures are very poor and not meaningful. They just simply illustrate that the numerical iterations reached a saturation value very early. But yields nothing more. I think it would be more insightful to present the payoff values in dependence of the model parameter instead of presenting a straightforward constant values in time.

The discussion part does not serve its original goal either. Actually it is not a proper discussion. Technical details should go to an appendix. Instead, it would be wiser to summarize the main findings here and compare them with earlier works. For instance the application of memory has a huge history in evolutionary game theory and related research already produced several interesting works, like Physica A 389 (2010) 2390-2396; Sci. Rep. 9 (2019) 262; Chaos, Solitons and Fractals 130 (2020) 109447. A related comment and a brief discussion about the general role of memory would be really useful.

I strongly believe that by addressing the critical points I raised will result in a significantly improved presentation which would expect a better response from the scientific community. Because the present version is very far the level that is expected from a paper to be published in this nice journal.

Decision letter (RSOS-202186.R0)

Dear Dr Ueda

The Editors assigned to your paper RSOS-202186 "Memory-two zero-determinant strategies in repeated games" have now received comments from reviewers and would like you to revise the paper in accordance with the reviewer comments and any comments from the Editors. Please note this decision does not guarantee eventual acceptance.

Please submit your revised manuscript and required files (see below) no later than 21 days from today's (ie 02-Mar-2021) date. Note: the ScholarOne system will 'lock' if submission of the

revision is attempted 21 or more days after the deadline. If you do not think you will be able to meet this deadline please contact the editorial office immediately.

on behalf of Professor Matjaz Perc (Associate Editor) and Miles Padgett (Subject Editor)
openscience@royalsociety.org

Reviewer comments to Author:
Reviewer: 1

Comments to the Author(s)

The author has investigated memory- n zero-determinant (ZD) strategies with general $n > 1$. The finding is that such strategies enforce linear relations between correlation functions of payoffs over n time steps. Overall, the manuscript is clearly written about an interesting topic, and such understanding has great potential importance if we consider the recent development of memory- n strategies that overcome shortcomings of the original ZD strategies. I did not follow all the details of the calculation, but the result looks convincing enough according to the numerical calculation. I recommend publication of this manuscript. Here are some minor comments that I would like the author to consider:

1. Line #32~33: "when the opponent 'unilaterally' defects twice."
2. If we think of the conventional prisoner's dilemma game, Eq. (3.6) seems over-determined because it contains 16 equations with 11 unknowns. Is it possible to specify conditions for the solution to exist?
3. Although Eq. (3.6) clearly shows an algebraic structure of the memory-two ZD strategies, I wonder if we really need the product of s_b and s_c on the right-hand side. There may be a degree of freedom to choose another function of s_b and s_c .

Reviewer: 2

Comments to the Author(s)

I would like to support this work because it contains solid calculation, but I strongly believe that present version is not appropriate for a general reader of R. Soc. Open Sci. Please note that this journal has highly interdisciplinary character with broad range of readers from rather different backgrounds. Therefore it is essential to present works in a way that is reachable for an ordinary reader.

For example the intro seems to be too brief and it is hard to find out what was really known earlier and what was the proper inspiration of present calculations. For example several previous works were ignored which also discussed extortion or TFT strategy in the mentioned social dilemma. Without giving a full list, let me just mention some representative papers: *Sci. Rep.* 4 (2014) 5496; *Nat. Commun.* 10 (2019) 783; *Chaos Solitons and Fractals* 141 (2020) 110421; *Phys. Rev. E* 80 (2009) 056104; *Chaos, Solitons and Fractals* 138 (2020) 109935.

It would be important to stress that present calculation assumes well-mixed populations while results may change in structured populations where interactions are fairly fixed and limited (see *J. R. Soc. Interface* 10 (2013) 20120997 and *EPL* 131 (2020) 68001).

In general I found that the figures are very poor and not meaningful. They just simply illustrate that the numerical iterations reached a saturation value very early. But yields nothing more. I think it would be more insightful to present the payoff values in dependence of the model parameter instead of presenting a straightforward constant values in time.

The discussion part does not serve its original goal either. Actually it is not a proper discussion. Technical details should go to an appendix. Instead, it would be wiser to summarize the main findings here and compare them with earlier works. For instance the application of memory has a huge history in evolutionary game theory and related research already produced several interesting works, like *Physica A* 389 (2010) 2390-2396; *Sci. Rep.* 9 (2019) 262; *Chaos, Solitons and Fractals* 130 (2020) 109447. A related comment and a brief discussion about the general role of memory would be really useful.

I strongly believe that by addressing the critical points I raised will result in a significantly improved presentation which would expect a better response from the scientific community. Because the present version is very far the level that is expected from a paper to be published in this nice journal.

===PREPARING YOUR MANUSCRIPT===

Your revised paper should include the changes requested by the referees and Editors of your manuscript. You should provide two versions of this manuscript and both versions must be provided in an editable format:
 one version identifying all the changes that have been made (for instance, in coloured highlight, in bold text, or tracked changes);
 a 'clean' version of the new manuscript that incorporates the changes made, but does not highlight them. This version will be used for typesetting if your manuscript is accepted.

===PREPARING YOUR REVISION IN SCHOLARONE===

-- If you have uploaded ESM files, please ensure you follow the guidance at <https://royalsociety.org/journals/authors/author-guidelines/#supplementary-material> to include a suitable title and informative caption. An example of appropriate titling and captioning may be found at https://figshare.com/articles/Table_S2_from_Is_there_a_trade-off_between_peak_performance_and_performance_breadth_across_temperatures_for_aerobic_sc_ope_in_teleost_fishes_/3843624.

Author's Response to Decision Letter for (RSOS-202186.R0)

See Appendix A.

RSOS-202186.R1 (Revision)

Review form: Reviewer 1

Is the manuscript scientifically sound in its present form?

Yes

Are the interpretations and conclusions justified by the results?

Yes

Is the language acceptable?

Yes

Do you have any ethical concerns with this paper?

No

Have you any concerns about statistical analyses in this paper?

No

Recommendation?

Accept as is

Comments to the Author(s)

The author's answers to my questions are all reasonable. I recommend publication.

Review form: Reviewer 2

Is the manuscript scientifically sound in its present form?

Yes

Are the interpretations and conclusions justified by the results?

Yes

Is the language acceptable?

Yes

Do you have any ethical concerns with this paper?

No

Have you any concerns about statistical analyses in this paper?

No

Recommendation?

Accept as is

Comments to the Author(s)

It is a significantly improved version. The author has addressed the critical points successfully, hence I am happy to support publication of the revised version.

Decision letter (RSOS-202186.R1)

Dear Dr Ueda,

It is a pleasure to accept your manuscript entitled "Memory-two zero-determinant strategies in repeated games" in its current form for publication in Royal Society Open Science. The comments of the reviewer(s) who reviewed your manuscript are included at the foot of this letter.

Please see the Royal Society Publishing guidance on how you may share your accepted author manuscript at <https://royalsociety.org/journals/ethics-policies/media-embargo/>. After publication, some additional ways to effectively promote your article can also be found here

<https://royalsociety.org/blog/2020/07/promoting-your-latest-paper-and-tracking-your-results/>.

on behalf of Miles Padgett (Subject Editor)
openscience@royalsociety.org

Reviewer comments to Author:

Reviewer: 2

Comments to the Author(s)

It is a significantly improved version. The author has addressed the critical points successfully, hence I am happy to support publication of the revised version.

Reviewer: 1

Comments to the Author(s)

The author's answers to my questions are all reasonable. I recommend publication.

Appendix A

Dear Editor,

Thank you very much for considering our manuscript. We hereby submit the revised manuscript and a letter of response to the reviewers' comments. In this submission, we provide two versions of the identical manuscript, with and without all revisions highlighted. They are submitted as separate files.

Sincerely yours,

Masahiko Ueda

Reply to the comments of Reviewer 1

We thank Reviewer 1 for comments. We reply to them.

[Referee's comment:]

1. Line #32-33: "when the opponent 'unilaterally' defects twice."

[Our reply:] In our revised version, we have modified the expression according to the suggestion of the referee.

[Referee's comment:]

2. If we think of the conventional prisoner's dilemma game, Eq. (3.6) seems overdetermined because it contains 16 equations with 11 unknowns. Is it possible to specify conditions for the solution to exist?

[Our reply:] It is difficult to specify the conditions for the solution of Eq. (3.6) to exist. Because the number of the components of a Press-Dyson (PD) matrix is M^{2N} and the number of payoff tensors $s_b \otimes s_c$ in Eq. (3.6) is $(N+1)^2$, most of memory-two strategies are not memory-two zero-determinant (ZD) strategies. However, the existence of ZD strategies is highly dependent on the structure of games, as in memory-one ZD strategies [30]. In our revised version, we have explained this fact below Eq. (3.6).

[Referee's comment:]

3. Although Eq. (3.6) clearly shows an algebraic structure of the memory-two ZD strategies, I wonder if we really need the product of s_b and s_c on the right-hand side. There may be a degree of freedom to choose another function of s_b and s_c .

[Our reply:] Yes, such choice instead of the product of s_b and s_c is possible. For example, in Ref. [49], memory-one ZD strategies where s_b is replaced by s_b^k were investigated. In our

revised version, we have explained this fact below Eq. (3.6) and cited Ref. [49].

Reply to the comments of Reviewer 2

We thank Reviewer 2 for comments. All of the comments were useful for improving presentation of our manuscript. We reply to them.

[Referee's comment:]

For example the intro seems to be too brief and it is hard to find out what was really known earlier and what was the proper inspiration of present calculations. For example several previous works were ignored which also discussed extortion or TFT strategy in the mentioned social dilemma. Without giving a full list, let me just mention some representative papers: *Sci. Rep.* 4 (2014) 5496; *Nat. Commun.* 10 (2019) 783; *Chaos Solitons and Fractals* 141 (2020) 110421; *Phys. Rev. E* 80 (2009) 056104; *Chaos, Solitons and Fractals* 138 (2020) 109935.

[Our reply:] In our revised version, we have modified Introduction for readers from different backgrounds and cited several papers. For details of modification, please check our manuscript file with changes highlighted.

[Referee's comment:]

It would be important to stress that present calculation assumes well-mixed populations while results may change in structured populations where interactions are fairly fixed and limited (see *J. R. Soc. Interface* 10 (2013) 20120997 and *EPL* 131 (2020) 68001).

[Our reply:] In our revised version, we have discussed that behaviors of memory-two ZD strategies may change when we consider evolutionary games with structured populations, in Section 6.

[Referee's comment:]

In general I found that the figures are very poor and not meaningful. They just simply illustrate that the numerical iterations reached a saturation value very early. But yields nothing more. I think it would be more insightful to present the payoff values in dependence of the model parameter instead of presenting a straightforward constant values in time.

[Our reply:] In our revised version, we have introduced a parameter q to the strategy of player 2 in Eq. (4.5) and changed q in the range $[0, 1]$ in our numerical simulations. Accordingly the horizontal axes of Figures 1-5 have been changed to q . We can check that linear relations hold for all q .

[Referee's comment:]

The discussion part does not serve its original goal either. Actually it is not a proper discussion. Technical details should go to an appendix. Instead, it would be wiser to summarize the main findings here and compare them with earlier works. For instance the application of memory has a huge history in evolutionary game theory and related research already produced several interesting works, like *Physica A* 389 (2010) 2390-2396; *Sci. Rep.* 9 (2019) 262; *Chaos, Solitons and Fractals* 130 (2020) 109447. A related comment and a brief discussion about the general role of memory would be really useful.

[Our reply:] In our revised version, we have summarized our main findings and discussed roles of memory in Section 6. Technical details about memory- n ZD strategies have been moved to Section 5.